# Assessment of health-related quality of life among Afghan refugees in Quetta, Pakistan

**Shoaib Kaleem[1]⊙, Tawseef Ahmad[2]⊙, Abdul Wahid[1]⊙, Hamad Haider Khan[3]‡, Tauqeer Hussain Mallhi[4]‡, Yaser Mohammed Al-Worafi[5]‡, Anila Alam[6]‡, Asad Khan[7]⊙*, Yusra Habib Khan[4]‡, Faiz Ullah Khan[ID][8]⊙ ***

1 Department of Pharmacy Practice, Faculty of Pharmacy and Health Sciences, University of Balochistan, Quetta, Pakistan, 2 Department of Clinical Pharmacy, Faculty of pharmaceutical sciences, Prince of Songkla University, Hat-Yai, Thailand, 3 Department of Endocrinology, First Affiliated Hospital of Xian Jiaotong University, Xi'an, China, 4 Department of Clinical Pharmacy, College of Pharmacy, Jouf University, Sakaka, Saudi Arabia, 5 College of Pharmacy, University of Science and Technology of Fujairah, Fujairah, United Arab Emirates, 6 Department of Pharmacy, Sardar Bahadur Khan Women University Balochistan, Quetta, Pakistan, 7 Discipline of Clinical Pharmacy, School of Pharmaceutical Sciences, Universiti Sains Malaysia, George Town, Penang, Malaysia, 8 Department of Pharmacy Administration and Clinical Pharmacy, School of Pharmacy, Xi'an Jiaotong University, Xi'an, China

⊙ These authors contributed equally to this work.
‡ HHK, THM, YMAW, AA, and YHK also contributed equally to this work.
* fkhan@bs.qau.edu.pk, faiz.rph@gmail.com (FUK); assad.pharmacist@gmail.com (AK)

**Data Availability Statement:** All relevant data are within the paper and its Supporting Information files.

## Abstract

The study aims to assess the health-related Quality of Life (HRQOL) and its association with socio-demographic factors among the Afghan refugees residing in Quetta, Pakistan. For this purpose, a cross-sectional, descriptive study design by adopting Euro QOL five dimensions questionnaire (EQ-5D) for the assessment of HRQOL was conducted by approaching Afghan refugees from the camp and other areas of Quetta, Pakistan. Furthermore, this study also involved descriptive analysis to expound participant's demographic characteristics while inferential statistics (Kruskal-Wallis and Mann–Whitney test, P < 0.05) were used to compare EQ-5D scale scores. All analyses were performed using SPSS v 20. Herein, a total of 729 participants were enrolled and were subsequently (n = 246, 33.7%) categorized based on their age of 22–31 years (31.30 ± 15.40). The results of mean EQ-5D descriptive score (0.85 ± 0.20) and EQ-VAS score (78.60 ± 11.10) indicated better HRQOL in the current study respondents as compared to studies conducted in other refugee camps around the globe. In addition, demographic characteristics including age, marital status, locality, years of living as refugees, life as a refugee residing out of Pakistan, place of residence in Afghanistan, educational qualification, occupation, and arrested for crime were the statistically significant predictors ($P < 0.05$) of EQ-5D index scores. However, gender, living status, monthly income, preferred place of treatment were non-significant predictors ($P > 0.05$). The results of current study provided evidence for a model that correlated with participant's socio-demographic information and HRQOL. Moreover, this study also revealed a baseline assessment for the health status of Afghan refugees, interestingly, these results could be applied for improving HRQOL of the given participants. In conclusion, the HRQOL of Afghan refugees residing in Quetta, Pakistan can largely be improved by providing

**Funding:** No specific funds were received for the current study.

**Competing interests:** The authors have declared that no competing interests exist.

adequate healthcare facilities, education and employment opportunities, mental and social support, and providing adequate housing and basic necessities of life.

## Introduction

A refugee is a person fleeing life-threatening conditions. In daily parlance and for journalistic purposes, this is roughly the meaning of "refugee hood." Predictably, in legal and political circles, among those officials who formulate refugee policies for states and international agencies, the meaning is considerably more circumscribed [1]. Worldwide, the number of refugees and asylum seekers is estimated to be about 11.5 million. Plus a much larger number of former refugees who have obtained a residence permit in a new country [2]. Among migrants fleeing war-torn areas, potentially traumatic events like close-quarters combat are frequent [3, 4], and many refugees may have mental illness as a result of these events [3, 5, 6] Everyone has the right to seek and receive asylum from persecution in other countries [7]. International assistance to refugees is channeled through United Nations High Commissioner for Refugees (UNHCR), through Non-Governmental Organizations (NGOs) and bilaterally. UNHCR is one of the few UN agencies that depends almost entirely on voluntary contributions to finance its operations. Less than two percent of UNHCR's annual budget comes from the United Nations; the rest is contributed by states, individuals, and the private sector [8]. Pakistan is neither a party to the 1951 Convention relating to the Status of Refugees nor the 1967 Protocol relating to the Status of Refugees [9]. The cost of refugee relief to Pakistan in the mid-1980s was about 1 million dollars a day, all of which was financed by contributions from foreign governments (channeled through UNHCR) and private voluntary organizations [10]. More than $273 million in humanitarian aid has already been given by the US to Pakistan's host communities and Afghan refugees there. The US donated approximately $60 million to the Pakistani host communities and Afghan refugees for the fiscal year 2022 [11]. Afghans in Pakistan are in some measure both refugees and economic migrants; they may face political threats in Afghanistan but are also working to support relatives there [12]. Migration and subsequent adjustment cause stress, which increases the risk of sickness [13]. Afghans comprise the largest refugee population in the world [14]. In general, social networks and support are seen as significant protective factors for both physical and mental health [15]. Pakistan has received Afghan refugees since the 1980s. There are 18 camps for Afghan refugees located in five districts of Balochistan Province. Six of the 18 camps are labeled as "new camps" and were established in response to the refugee influx in November 2001 [16]. The remaining 12 "old camps" cater to Afghan refugees who have been arriving since the time of the Russian invasion of Afghanistan in 1979. The number of Afghans in Pakistan is 3,049,268 individuals, as enumerated in the Census of Afghans in Pakistan 2005. Almost half reside in five districts (20.1 percent in Peshawar, 11.1 percent in Quetta, 7.6% in Nowshera, 5.1% in Pishin, and 4.3% in Karachi), while the balance is spread across the remaining 120 districts of Pakistan. According to the Census, there are 769,268 (25.2 percent) Afghans, or 115,565 families, in Balochistan province. The 2005 Census notes that if Pakistan's population ratios have not changed since the 1998 Census, then Afghans in Balochistan would be equivalent to 10 percent of Pakistanis, and in the Northwest Frontier Province (NWFP) the figure would be 7.6 percent [17]. Following the American invasion of Afghanistan, the number of refugees has increased in recent years. [18]. While in Pushtun-dominated areas, Afghan Pashtuns were allowed to settle [10]. Quetta district ranks second in Pakistan among the top ten districts with Afghans under the age of five (11.6

percent) [17]. In the last few decades, there has been an enormous amount of research on quality of life (QOL). Providing the exact meaning of the term "QOL" is rather difficult because an elaborate theoretical framework is (still) lacking. Therefore, there are nearly as many definitions of QOL as there are studies on this subject [19]. Research suggests that poor perceived present quality of life (QOL) may be the most significant factor in psychological illness and stress related disorders in refugee populations [20].

Health-related quality of life (HRQOL) is a multidimensional concept that focuses on the impact health status has on quality of life and represents the subjective evaluation of physical, mental, emotional, and social functioning [21]. Other studies tried to analyze the quality of life of the immigrants, often using other questionnaires as a tool to measure it [22]. As we know, in a lot of studies concerning medicinal interventions, QOL has become an important endpoint. There is a connection between health and quality of life, i.e., better health means a better quality of life. The current study reveals good HRQOL among Afghan refugees. To the best of our knowledge, this study is the very first to assess the HRQOL of Afghan refugees in Pakistan.

## Methodology

### Study setting and design

The study design was a population-based "cross-sectional study" using a quantitative approach. The research was being carried out in Quetta, Pakistan. Some Afghan refugees were living in random public places such as "Killi Kamalo, Zamindar Colony, and Bashir Chowk," called secondary research sites, but the majority of Afghan refugees were found in "Hazara Town, Jungle Bagh, Ghausabad, and Saranan Camp," called main research sites. As a reliable sample, the majority of the data was collected from the main research sites, and some was collected randomly from secondary research sites.

**Study sampling and study duration.** The duration of the study was from March 2017 to September 2017. A time-based study was being carried out, and a convenient sampling was done. The data collected was reliable because it was collected only from those who agreed to participate in the study as were willing to fill out the questionnaire. Those who did not agree to participate were excluded from the study.

**Study tool and statistical analysis.** The EQ5D-3L questionnaire along with demographic characteristics were used (S1 File). All the participants were asked to fill out the EQ5D-3L questionnaire. The EQ5D-3L (Urdu language) questionnaire was used. It was a self-administered questionnaire, but due to a lack of education, many of the samples were helped out to fill the questionnaire. All three versions are available in a wide range of languages and modes of administration [23]. The EQ-5D description system, which consists of five dimensions of health—mobility, self-care, common activities, pain/discomfort, and anxiety/depression—is also included. Each of these categories has three possible answers. Within a specific EQ5D dimension, the response tracks three severity levels (no problems/some or moderate problems/extreme problems).

*Scoring the EQ-5D-3 L VAS.* The EQ scale of the visual analog scale (VAS) measures subjects' self-reported health on a linear visual analog (vas scale ranging from 0 to 100, with terminals captioned 'The best conditions you could envision' and 'The very worst health you can think. precisely, on the EQ- visual analog scale (VAS), "100" represents the finest health level in addition to it the "0" represents the most severe health situation possible.

The version we used was EQ-5D-3L because it was available in translated Urdu. The collected data was analyzed and interpreted by using SPSS (Statistical Package for Social Sciences) 22.0 version software [24], the results obtained were shown in the form of frequency tables. The descriptive statistical analysis to expound participants' demographic characteristics and

the inferential statistics (Kruskal-Wallis and Mann-Whitney test, P < 0.05) were used to compare EQ-5D scale scores.

The ethical approval was obtained from the research committee of the department of pharmacy practice, faculty of pharmacy and health sciences, with reference number DPP/SS/83/17, University of Baluchistan, Quetta, Pakistan. According to the ethical guidelines, the study participants provided informed consent (S1 File). Additionally, we also obtained consent from the parents or guardians of the participants (only in the case of minors' participants).

## Results

The study sample consisted of 729 participants who took part in the study. Participants' ages ranged from 12 to 98 years. Here the Table 1 reflects the demographic information of the participants studied. Majority of the respondents were from 21 to 31 year of age and the dominated 480 (65.8%) participated respondent were male, 450 (61.7%) participants were married with 441 (60.5%) had urban residency in Afghanistan. The majority of participants (90.1%) (n = 657) did not reside in any nation other than Pakistan, with 234 (32.1%) of them having lived there for at least ten years. The majority of respondents (n = 222) were from Kandahar, and 95.5% of the 696 participants lived with their families. Almost 42% (n = 306) had no education, and 33.7% (n = 246) had a private job. 294 (40.3%) had no monthly income, and 79.4% (n = 579) preferred to be treated in a hospital or Basic health unit (BHU). Most of the respondent were not arrested for a crime. Mobility (n = 654), self-care (n = 666) (91.4%), usual work (n = 594) (81.5%), pain and discomfort (n = 519) (71.2%), anxiety and depression (n = 450) (61.7%) were all indicated as not being a problem by the majority of respondents by using self-reporting EQ-5D as shown in Table 2. A total of 24 health conditions as shown in Table 3, the majority of respondents (55.6%) have 1111 health states and no health issues. The EQ-5D Health Index/Score is significantly associated with age (p = 0.001), marital status (p = 0.001), locality in Afghanistan (p = 0.008), year living as refugees (p = 0.001), living as a refugee somewhere other than Pakistan (p = 0.004), place of residence in Afghanistan (p = 0.018), education (p = 0.033), occupation (p = 0.001), and being arrested for crime (p = 0.005) as shown in Table 4. In Table 5, age (p = 0.001), marital status (p = 0.001), education (p = 0.003), occupation (p = 0.001), and arrest for crime (p = 0.041) are significantly associated with VAS score.

## Discussion

The current study reveals good HRQOL among Afghan refugees. Keeping in mind that the descriptive score was a little bit higher than the perceived health status, this indicates that the actual health condition is better than what was perceived by the participants. To the best of our knowledge, this study is the very first to access the HRQoL of Afghan refugees in Pakistan The HRQOL of the relatively young adult Palestinian refugee population was relatively good for physical health, psychological health, and social relations, but was poor for environment, as the study was conducted in Jordan about the health-related quality of life of Palestinian refugees [21].The finding were again supported by the study QOL among Syrian refugees in Kurdistan which falls largely within a range of QOL scores [20]. This is consistent with earlier studies of the general population [15], immigrants in Sweden [25], torture survivors among refugees in Denmark [26], and a recently released study among Syrian refugees [27]. In contrast the Northern Greece refugees showed poor HRQL [28].

The current study was done with the EQ-5D instrument, which has five dimensions to determine the health status of the participants, including mobility, self-care, usual activities, pain or discomfort, and anxiety or depression. Each of the five dimensions is divided into three levels of problem perception. The current study included a total of 24 health statuses,

**Table 1. Demographic characteristics of study respondents (n = 729).**

| Description | Frequency | Percentage |
|---|---|---|
| **Age (years)** | | |
| 12 to 21 | 222 | 30.5 |
| 22 to 31 | 246 | 33.7 |
| 32 to 41 | 114 | 15.6 |
| 42 to 51 | 51 | 7.0 |
| 52 to 61 | 51 | 7.0 |
| 62 and above | 45 | 6.2 |
| **Gender** | | |
| Male | 480 | 65.8 |
| Female | 249 | 34.2 |
| **Marital Status** | | |
| Unmarried | 279 | 38.3 |
| Married | 450 | 61.7 |
| **Locality In Afghanistan** | | |
| Urban | 441 | 60.5 |
| Rural | 288 | 39.5 |
| **Year Living as Refugee in Pakistan** | | |
| 2 to 10 | 111 | 15.2 |
| 11 to 20 | 234 | 32.1 |
| 21 to 30 | 198 | 27.2 |
| 31 and above | 186 | 25.5 |
| **Live as Refugee other than Pakistan** | | |
| Yes | 72 | 9.9 |
| No | 657 | 90.1 |
| **Place of residence in Afghanistan** | | |
| Kabul | 66 | 9.1 |
| Jalalabad | 57 | 7.8 |
| Kandahar | 222 | 10.5 |
| Mazar Sharif | 63 | 8.6 |
| Herat | 36 | 4.9 |
| Sayyad/Sar-e-pol | 54 | 7.4 |
| Zabul | 30 | 4.1 |
| Baghlan | 30 | 4.1 |
| Helmand | 21 | 2.9 |
| Ghazni | 27 | 3.7 |
| Pashmul | 30 | 4.1 |
| Faryab | 18 | 2.5 |
| Kunduz | 75 | 10.3 |
| **Living Status** | | |
| Alone | 33 | 4.5 |
| With Family | 696 | 95.5 |
| **Education** | | |
| No Education | 306 | 42 |
| Religious | 129 | 17.7 |
| Primary | 72 | 9.9 |
| Middle | 66 | 9.1 |
| Matric | 39 | 5.3 |

(*Continued*)

**Table 1.** (Continued)

| Description | Frequency | Percentage |
|---|---|---|
| Inter | 45 | 6.2 |
| Graduation | 60 | 8.2 |
| Others | 12 | 1.6 |
| **Occupation** | | |
| Unemployed | 45 | 6.2 |
| Private Job | 246 | 33.7 |
| Own Business | 153 | 21 |
| Student | 87 | 11.9 |
| Housewife | 174 | 23.9 |
| Others | 24 | 3.3 |
| **Monthly Income** | | |
| No income | 294 | 40.3 |
| Less than 10,000 | 138 | 18.9 |
| 10,000 to 20,000 | 189 | 25.9 |
| 20,001 to 30,000 | 51 | 7.0 |
| More than 30,000 | 57 | 7.8 |
| **Preferred place of Treatment** | | |
| Hospital/BHU | 579 | 79.4 |
| General practitioner | 81 | 11.1 |
| Traditional healer | 69 | 9.5 |
| **Arrested for crime** | | |
| Yes | 39 | 5.3 |
| No | 690 | 94.7 |

**Table 2. Self-reported health status (EQ-5D).**

| EQ-5D Domain | Frequency | Percentage |
|---|---|---|
| **First Domain (Mobility)** | | |
| No Problem in walking about | 654 | 89.7 |
| Some Problem in Walking about | 75 | 10.3 |
| Confined to bed | 0 | 0 |
| **Second Domain (Self-care)** | | |
| No Problem in self-care | 666 | 91.4 |
| Some Problem in washing and dressing myself | 57 | 7.8 |
| Unable to wash and dress myself | 6 | 8 |
| **Third Domain (Usual Work)** | | |
| No Problem in performing usual activities | 594 | 81.5 |
| Some Problems in performing usual activities | 123 | 16.9 |
| Unable to perform usual activities | 12 | 1.6 |
| **Forth Domain (Pain and Discomfort)** | | |
| No pain and discomfort | 519 | 71.2 |
| Some pain and discomfort | 195 | 26.7 |
| Extreme pain and discomfort | 15 | 2.1 |
| **Fifth Domain (Anxiety and Depression)** | | |
| Not anxious or depress | 450 | 61.7 |
| Moderately anxious or depress | 267 | 36.6 |
| Extremely anxious or depress | 12 | 1.6 |

**Table 3. Frequency of self-reported (EQ-5D) health states.**

| S No | EQ-5D states | Frequency | Percentage |
|---|---|---|---|
| 1 | 11111 | 405 | 55.6 |
| 2 | 11112 | 90 | 12.3 |
| 3 | 11113 | 6 | 0.8 |
| 4 | 11121 | 12 | 1.6 |
| 5 | 11122 | 57 | 7.8 |
| 6 | 11132 | 3 | 0.4 |
| 7 | 11211 | 3 | 0.4 |
| 8 | 11221 | 6 | 0.8 |
| 9 | 11222 | 45 | 6.2 |
| 10 | 11311 | 3 | 0.4 |
| 11 | 11331 | 3 | 0.4 |
| 12 | 12111 | 3 | 0.4 |
| 13 | 12122 | 3 | 0.4 |
| 14 | 12211 | 3 | 0.4 |
| 15 | 12222 | 3 | 0.4 |
| 16 | 12231 | 3 | 0.4 |
| 17 | 13333 | 6 | 0.8 |
| 18 | 21112 | 3 | 0.4 |
| 19 | 21121 | 3 | 0.4 |
| 20 | 21122 | 9 | 1.2 |
| 21 | 21211 | 3 | 0.4 |
| 22 | 21221 | 3 | 0.4 |
| 23 | 21222 | 12 | 1.6 |
| 24 | 22222 | 42 | 5.8 |

with the most dominant EQ-5D status having no problems in all five domains. The current study findings are also in line with what is reported in a study done in the Netherlands, which reported that Afghan refugees considered their physical and mental health to be good [2]. In comparison to the current study in the fifth domain, where the majority of respondents were not depressed, two different studies elsewhere in the world reported that distress symptoms occur at higher rates in the US and Afghans residing in Turkey are at high risk of psychopathology [29, 30]. There are many factors that influence the health of Afghan refugees. As we know, resettlement in a new culture may lead to many health problems. Health may be affected by starting your life in a country with different culture, religion, language, social structure, values, and environment, as well as by the behavior of the people where refugees are resettled. The areas of Pakistan where Afghan refugees are living have the same culture and religion. As we know, the majority of the Afghans living in Quetta are Pashtuns [17], and the residing people in Quetta are also of the same ethnicity having same values, environment, and social structure- even the dressing is same. Pashtunistan, the land of the Pashtuns or Pathans, lies on both sides of the Durand Line, the frontier between Pakistan and Afghanistan. Since 1947, Afghanistan has consistently repudiated the Durand Line and demanded the right of self-determination for the Pashtuns because she does not consider them a part of Pakistan [31], and the Pashtun areas of Balochistan, including Quetta, also believe in the ideology of Pashtunistan and behave like brothers with Afghan refugees.

HRQOL had a significant relationship with many of the demographic characteristics in our study like age, marital status, locality in Afghanistan, living as a refugee somewhere other than

**Table 4. Different variables EQ-5D Score (0.85 ± 0.20).**

| Description | Frequency | Mean ± SD | P value |
|---|---|---|---|
| **Age\* (years)** | | | |
| 12 to 21 | 222 | 0.95 ± 0.10 | **0.001** |
| 22 to 31 | 246 | 0.86 ± 0.23 | |
| 32 to 41 | 114 | 0.90 ± 0.12 | |
| 42 to 51 | 51 | 0.77 ± 0.19 | |
| 52 to 61 | 51 | 0.16 ± 0.15 | |
| 62 and above | 45 | 0.60 ± 0.29 | |
| **Gender\*\*** | | | |
| Male | 480 | 0.86 ± 0.22 | 0.082 |
| Female | 249 | 0.84 ± 0.18 | |
| **Marital Status\*\*** | | | |
| Unmarried | 279 | 0.89 ± 0.24 | **0.001** |
| Married | 450 | 0.83 ± 0.17 | |
| **Locality in Afghanistan\*\*** | | | |
| Urban | 441 | 0.87 ± 0.20 | **0.008** |
| Rural | 288 | 0.82 ± 0.21 | |
| **Year Living as Refugee\*** | | | |
| 2 to 10 | 111 | 0.93 ± 0.21 | **0.001** |
| 11 to 20 | 234 | 0.87 ± 0.17 | |
| 21 to 30 | 198 | 0.86 ± 0.17 | |
| 31 and above | 186 | 0.77 ± 0.25 | |
| **Live As Refugee other than Pakistan\*\*** | | | |
| Yes | 72 | 0.69 ± 0.34 | **0.004** |
| No | 657 | 0.87 ± 0.18 | |
| **Place of residence in Afghanistan\*** | | | |
| Kabul | 66 | 0.72 ± 0.35 | **0.018** |
| Jalalabad | 57 | 0.91 ± 0.13 | |
| Kandahar | 222 | 0.89 ± 0.16 | |
| Mazar Sharif | 63 | 0.88 ± 0.15 | |
| Herat | 36 | 0.90 ± 0.12 | |
| Sayyad/Sar-e-pol | 54 | 0.92 ± 0.13 | |
| Zabul | 30 | 0.81 ± 0.29 | |
| Baghlan | 30 | 0.69 ± 0.17 | |
| Helmand | 21 | 0.84 ± 0.12 | |
| Ghazni | 27 | 0.78 ± .25 | |
| Pashmul | 30 | 0.94 ± .12 | |
| Faryab | 18 | 0.92 ± .11 | |
| Kunduz | 75 | 0.85 ± .26 | |
| **Living Status\*\*** | | | |
| Alone | 33 | 0.68 ± 0.40 | 0.083 |
| With Family | 696 | 0.86 ± 0.19 | |
| **Education\*** | | | |
| No Education | 306 | 0.84 ± 0.16 | **0.033** |
| Religious | 129 | 0.81 ± 0.20 | |
| Primary | 72 | 0.89 ± 0.14 | |
| Middle | 66 | 0.89 ± 0.28 | |
| Matric | 39 | 0.87 ± 0.17 | |
| Inter | 45 | 0.84 ± 0.26 | |
| Graduation | 60 | 0.96 ± 0.06 | |
| Others | 12 | 0.60 ± 0.60 | |

*(Continued)*

**Table 4.** (Continued)

| Description | Frequency | Mean ± SD | P value |
|---|---|---|---|
| **Occupation*** | | | |
| Unemployed | 45 | 0.79 ± 0.20 | **0.001** |
| Private Job | 246 | 0.89 ± 0.18 | |
| Own Business | 153 | 0.80 ± 0.26 | |
| Student | 87 | 0.97 ± 0.06 | |
| Housewife | 174 | 0.82 ± 0.16 | |
| Others | 24 | 0.66 ± 0.36 | |
| **Monthly Income*** | | | |
| No income | 294 | 0.84 ± 0.19 | 0.095 |
| Less than 10,000 | 138 | 0.88 ± 0.29 | |
| 10,000 to 20,000 | 189 | 0.85 ± 0.19 | |
| 20,001 to 30,000 | 51 | 0.84 ± 0.15 | |
| More than 30,000 | 57 | 0.88 ± 0.14 | |
| **Preferred place of Treatment*** | | | |
| Hospital/BHU | 579 | 0.86 ± 0.21 | 0.372 |
| General practitioner | 81 | 0.82 ± 0.21 | |
| Traditional healer | 69 | 0.84 ± 0.17 | |
| **Arrested for crime**** | | | |
| Yes | 39 | 0.62 ± 0.42 | **0.005** |
| No | 690 | 0.87 ± 0.18 | |

* Kruskal Wallis Test.

** Mann Whitney Test.

**Table 5. VAS Score (78.60 ± 11.10) among variables.**

| Description | Frequency | Mean ± SD | P value |
|---|---|---|---|
| **Age* (years)** | | | |
| 12 to 21 | 222 | 83.45 ± 7.62 | **0.001** |
| 22 to 31 | 246 | 80.00 + 10.83 | |
| 32 to 41 | 114 | 77.76 + 9.56 | |
| 42 to 51 | 51 | 72.65 + 16.01 | |
| 52 to 61 | 51 | 70.00 + 9.84 | |
| 62 and above | 45 | 65.67 + 7.28 | |
| **Gender**** | | | |
| Male | 480 | 78.84 + 10.89 | 0.597 |
| Female | 249 | 78.13 + 11.54 | |
| **Marital Status**** | | | |
| Unmarried | 279 | 82.10 + 10.74 | **0.001** |
| Married | 450 | 76.43 + 10.79 | |
| **Locality In Afghanistan**** | | | |
| Urban | 441 | 79.73 + 10.73 | 0.057 |
| Rural | 288 | 76.88 + 11.47 | |
| **Year Living as Refugee*** | | | |
| 2 to 10 | 111 | 81.89 + 7.84 | 0.074 |
| 11 to 20 | 234 | 79.10 + 9.89 | |
| 21 to 30 | 198 | 78.86 + 11.56 | |
| 31 and above | 186 | 75.73 + 13.08 | |

(*Continued*)

**Table 5.** (Continued)

| Description | Frequency | Mean ± SD | P value |
|---|---|---|---|
| **Live As Refugee other than Pakistan**** | | | |
| Yes | 72 | 80.63 + 11.45 | 0.320 |
| No | 657 | 78.38 + 11.06 | |
| **Place of residence in Afghanistan*** | | | |
| Kabul | 66 | 75.68 + 12.27 | 0.323 |
| Jalalabad | 57 | 81.32 + 10.78 | |
| Kandahar | 222 | 78.85 + 10.74 | |
| Mazar Sharif | 63 | 76.19 + 7.73 | |
| Herat | 36 | 78.75 + 8.01 | |
| Sayyad/Sar-e-pol | 54 | 78.89 + 8.14 | |
| Zabul | 30 | 80.00 + 12.69 | |
| Baghlan | 30 | 70.00 + 14.72 | |
| Helmand | 21 | 76.43 + 12.48 | |
| Ghazni | 27 | 85.00 + 8.66 | |
| Pashmul | 30 | 82.00 + 10.59 | |
| Faryab | 18 | 80.93 + 10.68 | |
| Kunduz | 75 | 79.40 + 14.16 | |
| **Living Status**** | | | |
| Alone | 33 | 79.55 + 13.68 | 0.569 |
| With Family | 696 | 78.56 + 10.98 | |
| **Education*** | | | |
| No Education | 306 | 76.52 + 10.11 | **0.003** |
| Religious | 129 | 78.72 + 10.86 | |
| Primary | 72 | 79.58 + 9.88 | |
| Middle | 66 | 82.73 + 10.88 | |
| Matric | 39 | 78.85 + 12.10 | |
| Inter | 45 | 77.33 + 13.47 | |
| Graduation | 60 | 86.00 + 11.53 | |
| Others | 12 | 68.75 + 1436 | |
| **Occupation*** | | | |
| Unemployed | 45 | 74.33 + 13.99 | **0.001** |
| Private Job | 246 | 79.02 + 9.07 | |
| Own Business | 153 | 78.73 + 12.48 | |
| Student | 87 | 86.03 + 8.27 | |
| Housewife | 174 | 76.03 + 10.95 | |
| Others | 24 | 73.13 + 11.10 | |
| **Monthly Income*** | | | |
| No income | 294 | 77.96 + 12.13 | 0.966 |
| Less than 10,000 | 138 | 79.57 + 9.59 | |
| 10,000 to 20,000 | 189 | 78.81 + 10.65 | |
| 20,001 to 30,000 | 51 | 79.41 + 9.66 | |
| More than 30,000 | 57 | 78.16 + 12.38 | |
| **Preferred place of Treatment*** | | | |
| Hospital/BHU | 579 | 79.30 + 10.41 | 0.089 |
| General practitioner | 81 | 77.41 + 14.36 | |
| Traditional healer | 69 | 74.13 + 11.74 | |
| **Arrested for crime**** | | | |

*(Continued)*

**Table 5.** (Continued)

| Description | Frequency | Mean ± SD | P value |
|---|---|---|---|
| Yes | 39 | 71.92 + 11.99 | **0.041** |
| No | 690 | 78.98 + 10.95 | |

\* Kruskal Wallis Test

\*\* Mann Whitney Test.

Pakistan, year of Living as a refugee, place of residence in Afghanistan, education, occupation and being arrested for crime. There are mixed results when our findings are compared with those of other studies of the same nature. A study conducted in Italy reported that age was influential variable on HRQOL among refugees and asylum seekers using the SF-36 tool [22]. A study done using WHOQOL-BREF reported that age is significantly associated with physical, social and environmental domains [32]. Gerritsen et al. found that age, marital status, education, and year as refugees in that country were all significant factors [2]. A study mentioned that immigration background, gender, and age are significantly associated with physical and mental health [33]. Demographic factors such as employment status and income were found to be significantly related to the mental health of Afghan refugees living in Istanbul [30]. Although the majority of the respondents were not arrested for any crime. Beside the crime, if someone has done something wrong, authorities should act against them, but targeting someone only because he is Afghan isn't fair [34]. It can be a reason that some of the respondents had problems in the psychological domain and may be age -dependent. A greater HRQOL is achieved by taking into account factors like similarity in culture, religion, language, social structure, values, and surroundings, as well as how people treat refugees.

## Study strengths and weaknesses

The majority of the refugee camps in the Quetta region were covered by the survey, which was done in the Balochistan province's capital city. which will assist the government and policymakers in improving the HRQOL of Afghan refugees living in these camps. However, because we employ convenience sampling, which may limit the study's generalizability, we are unable to generalize these findings to all Afghan refugees in the country as a whole.

## Conclusion

The current study reveals better HRQOL among refugees in Quetta, Pakistan. Results may vary elsewhere in Baloch areas and Panjab, Pakistan, due to changes in culture, language, social structure, values, and environmental conditions. Our study concluded that demographic characteristics like gender, living status, monthly income, and preferred place of treatment are not associated with health-related quality of life among Afghan refugees; however, the effect varies with age, marital status, locality in Afghanistan, year living as refugees, living as a refugee somewhere other than Pakistan, place of residence in Afghanistan, education, occupation, and being arrested for crime, which were significantly associated with the EQ-5D Health Index/ Score. This study provides a baseline assessment for the health status of Afghan refugees and the results could be applied in improving HRQOL. It can improve more by improving the educational status and environmental variable such as income, which can result in better HRQOL. Future research should attempt to address educational issues in relation to the quality of life of refugees, as the identification of educational issues may assist in the design of educational-specific interventions in self-sustenance and improving quality of life.

## Supporting information

**S1 File.**
(PDF)

## Acknowledgments

We would like to thank all the contributors who helped us in the present research project.

## Author Contributions

**Conceptualization:** Shoaib Kaleem, Tauqeer Hussain Mallhi.

**Data curation:** Abdul Wahid, Anila Alam, Yusra Habib Khan.

**Formal analysis:** Abdul Wahid.

**Funding acquisition:** Yaser Mohammed Al-Worafi.

**Investigation:** Yaser Mohammed Al-Worafi.

**Methodology:** Abdul Wahid, Anila Alam.

**Writing – review & editing:** Tawseef Ahmad, Hamad Haider Khan, Asad Khan, Faiz Ullah Khan.

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
