## [Decision Letter · Decision Letter 0]

28 Jan 2022

PONE-D-21-26924

Assessment of health-related quality of life among Afghan refugees in Quetta, Pakistan

PLOS ONE

Dear Dr. Khan,

Thank you for submitting your manuscript to PLOS ONE. After careful consideration, we feel that it has merit but does not fully meet PLOS ONE’s publication criteria as it currently stands. Therefore, we invite you to submit a revised version of the manuscript that addresses the points raised during the review process.

We look forward to receiving your revised manuscript.

Kind regards,

Dylan A Mordaunt, MB ChB, MPH, MHLM, FRACP, FAIDH

Academic Editor

PLOS ONE

https://journals.plos.org/plosone/s/file?id=ba62/PLOSOne_formatting_sample_title_authors_affiliations.pdf”.

A clean copy of the edited manuscript (uploaded as the new *manuscript* file).

3. In your Methods section, please provide a justification for the sample size used in your study, including any relevant power calculations (if applicable).

4. You indicated that you had ethical approval for your study. In your Methods section, please ensure you have also stated whether you obtained consent from parents or guardians of the minors included in the study or whether the research ethics committee or IRB specifically waived the need for their consent.

Furthermore  thank you for your statement: ethical approval was obtained from the research committee department of Pharmacy

Practice, Faculty of Pharmacy and Health Sciences with number (DPP/SS/83/17) University

of Baluchistan, Quetta, Pakistan

“Nil”

7. We note that you have indicated that data from this study are available upon request. PLOS only allows data to be available upon request if there are legal or ethical restrictions on sharing data publicly. For more information on unacceptable data access restrictions, please see http://journals.plos.org/plosone/s/data-availability#loc-unacceptable-data-access-restrictions.

8. Thank you for submitting the above manuscript to PLOS ONE. During our internal evaluation of the manuscript, we found significant text overlap between your submission and the following previously published works, some of which you are an author.

-https://www.coursehero.com/file/41355227/Refugees-of-Venezuelans-in-Colombiadocx/

-https://moam.info/refugee-protection_599bf6501723dd0f406edd9c.html

-https://journals.sagepub.com/doi/pdf/10.1177/019791839603000301

-https://academic.oup.com/jrs/article-abstract/21/1/43/1513931?redirectedFrom=fulltext

-https://www.ijidonline.com/article/S1201-9712(05)00169-4/fulltext

-https://www.livelihoods.org/hot_topics/docs/Afghans_Quetta.pdf

-https://linkinghub.elsevier.com/retrieve/pii/S0029655417300891

Please revise the manuscript to rephrase the duplicated text, cite your sources, and provide details as to how the current manuscript advances on previous work. Please note that further consideration is dependent on the submission of a manuscript that addresses these concerns about the overlap in text with published work.

Additional Editor Comments:

Thank you for your submission. This is an important and timely piece of work. Both reviewers have valuable suggestions but in taking on board reviewer 2's comment about timing, I have framed the revisions needed as minor and intend to progress this to acceptance assuming the issues are addressed. This doesn't discount reviewer 1's comments, which are important and should be addressed.

With specific reference to the criteria for publication:

1. The study appears to present the results of original research.

2. Results do not appear to have been published elsewhere. It would be helpful to clarify whether a preprint is already available and might be worth considering in order to make the work available early for reference.

3. Experiments, statistics, and other analyses are performed to an appropriate standard and are described in sufficient detail.

4. Conclusions are presented in an appropriate fashion and are supported by the data.

5. The article is presented in an intelligible fashion and is written in standard English- with a few comments noted by the reviewers.

6. The research meets all applicable standards for the ethics of experimentation and research integrity- an ethics approval document is included, but it would be helpful to include a line in text for ease of reference (in the methods).

7. The article adheres to appropriate reporting guidelines and community standards for data availability. Though this standard isn't directly applicable, it would be worth checking with the structure and ensuring all relevant content has been considered for inclusion- https://pubmed.ncbi.nlm.nih.gov/8973129/.

Reviewers' comments:

Reviewer's Responses to Questions

**Comments to the Author**

1. Is the manuscript technically sound, and do the data support the conclusions?

Reviewer #1: Partly

Reviewer #2: Yes

2. Has the statistical analysis been performed appropriately and rigorously? 

Reviewer #1: N/A

Reviewer #2: Yes

3. Have the authors made all data underlying the findings in their manuscript fully available?

Reviewer #1: Yes

Reviewer #2: Yes

4. Is the manuscript presented in an intelligible fashion and written in standard English?

Reviewer #1: No

Reviewer #2: No

5. Review Comments to the Author

Reviewer #1: Assessing health-related quality of life among Afghan refugees is an important topic.

1. The manuscript requires a thorough language check. Some examples include

- Variables other than proper nouns should be lower case (abstract, methods and results)

- "Afghan" should be capitalized

-Table # should be capitalized

-Consider omitting or rephrasing the first sentences of the introduction. The current phrasing is too colloquial, the definition of refugee should be referenced

2. The flow of the introduction is hard to follow in some parts and others could be omitted or included as a supplement e.g. the sentences about the old camps and exact distribution.

3. The HRQOL sentences should be moved to the methods

4. Discussion "The current study reveals better HRQOL.. " better than what? The comparison is missing

5. Add 5-10 more current references from the last 3-5 years.

6. Table titles please add the setting/participants and year of the study possibly, Tables 2-5 need a proper title, please check the journal for examples. EQ-5d should be spelled out and described in the notes of each table

7. Table 3 the meaning of columns S No and EQ-5D are unclear

Reviewer #2: This study by Kaleem et al analysed questionnaire based data to assess health-related quality of life amongst Afghan refugees in Pakistan. This manuscript is of particular importance given the current situation in Afghanistan.

To be accepted for publication the written English must be improved within the methods, results and discussion sections. The language is currently not at a high enough standard for publication. E.g. there are capital letter appearing in the middle of sentences when they are not warranted.

The formatting of the tables is a problem (rows don't align) and need to be re-formatted to make it easier to interpret. The tables are not well described in the titles, which seem to have random numbers inserted within the titles without any description.

6. PLOS authors have the option to publish the peer review history of their article (what does this mean?). If published, this will include your full peer review and any attached files.

Reviewer #1: No

Reviewer #2: No

---

## [Author Response · Author response to Decision Letter 0]

14 Feb 2023

Editor comments and Responses 

Comment 01

Response 

Dear Editor, the manuscript file has been edited as per the PLOS ONE format, as mentioned on the journal site.

Comment 02

We suggest you thoroughly copyedit your manuscript for language usage, spelling, and grammar.

Response 

Dear Editor, the manuscript has been thoroughly checked and modified as needed.

Comment 03 

In your Methods section, please provide a justification for the sample size used in your study, including any relevant power calculations (if applicable).

Response 

Dear editor, A time-based study was being carried out, and a convenient sampling was done. We are not aware of any kind of finite population or specific population of Afghan refugees in the community. That’s why we didn’t use any specific statistical formula for the finite population. But we ensured the maximum sample was as large as possible to include in the study within the study period.

Comment 04 

You indicated that you had ethical approval for your study. In your Methods section, please ensure you have also stated whether you obtained consent from parents or guardians of the minors included in the study or whether the research ethics committee or IRB specifically waived the need for their consent.

Response 

Dear Editor, we have received ethical approval from the Research Ethical Committee at the University of Balochistan. Furthermore, we also obtained verbal consent from the participants to follow research ethics. The ethical committee's approval has been attached as a supplementary file.

Comment 06

In your Data Availability statement, you have not specified where the minimal data set underlying the results described in your manuscript can be found.

Response 06

All the data are included in the manuscript. Additionally, the questionnaire used for the study is also included in the supplementary files. The raw data and before-analysis files are available on reasonable request to the corresponding author.

Comment 07 

We note that you have indicated that data from this study are available upon request. PLOS only allows data to be available upon request if there are legal or ethical restrictions on sharing data publicly.

Response 07 

We mentioned the above statement only for the raw data; the rest of the summaries and questionnaire are freely available.

Comment 08

Thank you for submitting the above manuscript to PLOS ONE. During our internal evaluation of the manuscript, we found significant text overlap between your submission and the following previously published works, some of which you are an author.

Response 08

The manuscript has been revised and rephrased to avoid overlapping.

Reviewer 01 Comments 

Comment 01

The manuscript requires a thorough language check. Some examples include

- Variables other than proper nouns should be lower case (abstract, methods and results)

- "Afghan" should be capitalized

-Table # should be capitalized

Response 01

The manuscript has been rigorously checked for language, spelling, and grammar corrections. All the variables other than nouns have been changed to lower case. The word tables has been capitalized. 

Comment 02

The flow of the introduction is hard to follow in some parts and others could be omitted or included as a supplement e.g., the sentences about the old camps and exact distribution.

Response 

We rephrased the complex sentence in the introduction into simple sentences for better understanding.

Comments 03

The HRQOL sentences should be moved to the methods. 

Response 

The sentence has been moved to the methodology part.

Comment 04

Discussion "The current study reveals better HRQOL.. " better than what? The comparison is missing. 

Response 

The current study reveals that Afghan refugees have better HRQOL because they share the same religions, ethnicity, social and environmental values. 

Comment 05

Add 5-10 more current references from the last 3-5 years.

Response 

We added nine more recent references, ranging from 2016 to 2022.

Comment 06

Table titles please add the setting/participants and year of the study possibly, Tables 2-5 need a proper title, please check the journal for examples. EQ-5d should be spelled out and described in the notes of each table

Response 

The table title has been changed. 

Comment 07

Table 3 the meaning of columns S No and EQ-5D are unclear

Response

EQ-5D represents the frequency of health states; we have now modified it for a better understanding. 

Reviewer 02 Comments 

To be accepted for publication the written English must be improved within the methods, results and discussion sections. The language is currently not at a high enough standard for publication. E.g. there are capital letter appearing in the middle of sentences when they are not warranted.

The formatting of the tables is a problem (rows don't align) and need to be re-formatted to make it easier to interpret. The tables are not well described in the titles, which seem to have random numbers inserted within the titles without any description.

Response 

The manuscript has been checked thoroughly, and capital letters have been removed. The format of the tables changed as per suggestions.

---

## [Decision Letter · Decision Letter 1]

3 Apr 2023

PONE-D-21-26924R1Assessment of health-related quality of life among Afghan refugees in Quetta, PakistanPLOS ONE

Dear Dr. Faiz Ullah Khan,

Thank you for submitting your manuscript to PLOS ONE. After careful consideration, we feel that it has merit but does not fully meet PLOS ONE’s publication criteria as it currently stands. Therefore, we invite you to submit a revised version of the manuscript that addresses the points raised during the review process.

 Please follow the reviewer's comments and submit again if you are still interested to.

We look forward to receiving your revised manuscript.

Kind regards,

AKM Alamgir, PhD

Academic Editor

PLOS ONE

Additional Editor Comments (if provided):

Please revise as suggested.

Reviewers' comments:

Reviewer's Responses to Questions

**Comments to the Author**

1. If the authors have adequately addressed your comments raised in a previous round of review and you feel that this manuscript is now acceptable for publication, you may indicate that here to bypass the “Comments to the Author” section, enter your conflict of interest statement in the “Confidential to Editor” section, and submit your "Accept" recommendation.

Reviewer #2: (No Response)

Reviewer #3: All comments have been addressed

2. Is the manuscript technically sound, and do the data support the conclusions?

Reviewer #2: Yes

Reviewer #3: Yes

3. Has the statistical analysis been performed appropriately and rigorously? 

Reviewer #2: Yes

Reviewer #3: No

4. Have the authors made all data underlying the findings in their manuscript fully available?

Reviewer #2: No

Reviewer #3: Yes

5. Is the manuscript presented in an intelligible fashion and written in standard English?

Reviewer #2: Yes

Reviewer #3: No

6. Review Comments to the Author

Reviewer #2: I thank the authors for this resubmission which is much improved. I do however have some comments to be addressed in a revised submission.

1) There is a typo in the list of authors.

Abstract:

2) The authors state in the abstract that the 729 participants were randomly enrolled. What methodology was used for randomization?

3) The authors state that this study indicates better HRQOL in the current study respondents. Better compared to what/who?

4) Although increasing monthly income is important for HRQOL, it was not a predictor of quality of EQ-5D index scores in this cohort. Therefore, does this not indicate that this is not an important factor in this cohort's EQ-5D. As such, it may be premature to suggest in the conclusions that uplifting monthly income would improve HRQOL.

Introduction:

5) Spell out UNHCR and NGO.

6) Why have the authors only included costings from mid 1980s, which is approx. 40 years out of date? It is ok to include historical costings in relation to historical resettlement, but should this should be brought more current. i.e also including recent figures.

7) QOL is defined twice within the introduction - only needs to be done in the first instance.

8) Typo - used QoL instead of QOL in last paragraph of the introduction.

Tables:

9) There are no legends, which makes it difficult to understand the tables. e.g. What is Frequency (243) in table 1, what is [mean+/-SD] referring to in table 1? Legends need to be included for each of the tables and not just a title.

10) Table 1 - "Year Living as Refugee" is mis-aligned in table 1.

Methods:

11) Further description of the scoring system(s) are required within the methods section.

Results:

11) The authors state that the majority of participants were from Kandahar. This is incorrect, Kandahar was the largest single recruitment site.

12) Define BHU.

13) should be "Table 2" not "Table 02".

14) Most of the results are just a description of the study population. Needs further reporting on the findings of the EQ-5D data and VAS score.

15) The first time VAS score is introduced is the last sentence within the results. It is not described anywhere else prior to this. Needs to be included in the methodology. This data is lacking background and context.

Discussion:

16) Some points of the scoring system(s) within the discussion need to be introduced earlier (Methodology/results). This will allow the reader to fully understand the results and what they mean before reaching the discussion.

Reviewer #3: • It is recommended to perform multivariate analysis (regression).

• Is the Urdu version of the EQ5D-3Lquestionnaire valid and reliable? Please provide a reference. And do all Afghan refugees in Pakistan have the ability to understand Urdu? Did you get the data through interviews?

• In the method field, write the name of the analysis used.

• In the discussion section, write the strengths and weaknesses of the study.

• And keep in mind that your study does not have external validity because sampling was available and cannot be generalized to the Afghan refugee community in Pakistan.

7. PLOS authors have the option to publish the peer review history of their article (what does this mean?). If published, this will include your full peer review and any attached files.

Reviewer #2: No

Reviewer #3: No

---

## [Author Response · Author response to Decision Letter 1]

3 May 2023

Reviewer 2 comments and Response from authors 

Comment: 

There is a typo in the list of authors.

Response: 

Thanks, now we rewrite the names of authors as per the template of PLOS ONE. 

Comment

Abstract:

2) The authors state in the abstract that the 729 participants were randomly enrolled. What methodology was used for randomization?

Response: 

Thanks! To be able to collect data to study sample populations, we use convenience sampling approaches. according to our description in the methods section.

Comment

3) The authors state that this study indicates better HRQOL in the current study respondents. Better compared to what/who?

Response: 

Thanks! Now that it has been re-written in a clear way, our study findings indicate better HRQOL as compared to other refugee HRQOL studies.

Comment

4) Although increasing monthly income is important for HRQOL, it was not a predictor of quality of EQ-5D index scores in this cohort. Therefore, does this not indicate that this is not an important factor in this cohort's EQ-5D. As such, it may be premature to suggest in the conclusions that uplifting monthly income would improve HRQOL.

Response: 

Thanks! For a better understanding and suggestions on how to enhance the HRQOL of refugees, we have revised the conclusion.

Comment

Introduction:

5) Spell out UNHCR and NGO.

Response: 

Thanks, we mentioned the complete abbreviation of UNHCR and NGO in the manuscript. 

Comment

6) Why have the authors only included costings from mid 1980s, which is approx. 40 years out of date? It is ok to include historical costings in relation to historical resettlement, but should this should be brought more current. i.e also including recent figures.

Response: 

Thanks, we have been updated now with the cost as per the figure from recently.

Comment

7) QOL is defined twice within the introduction - only needs to be done in the first instance.

Response: 

Thanks now, it's changed to only once.

Comment

8) Typo - used QoL instead of QOL in last paragraph of the introduction.

Response: 

Thanks, corrected now 

Comment

Tables:

9) There are no legends, which makes it difficult to understand the tables. e.g. What is Frequency (243) in table 1, what is [mean+/-SD] referring to in table 1? Legends need to be included for each of the tables and not just a title. 10) Table 1 - "Year Living as Refugee" is mis-aligned in table 1.

Response: 

Thanks! We have now improved the table in a clearer way for easy understanding.

Comment

Methods:

11) Further description of the scoring system(s) are required within the methods section.

Response: 

Thanks, the scoring description has been mentioned now. 

Comment

Results:

11) The authors state that the majority of participants were from Kandahar. This is incorrect, Kandahar was the largest single recruitment site.

Response: 

Thanks, Kandahar wasn’t the recruitment site for us, but the majority of respondents or the population in our study were migrated from Kandahar, and because Kandahar is the nearest province of Afghanistan to the Chaman border and Quetta, this is the reason the majority of refugees in the study population were from Kandahar.

Comment

12) Define BHU.

Response: 

Thank you! we referred to as a basic health unit.

Comment

13) should be "Table 2" not "Table 02".

Response: 

Thanks, corrected 

Comment

14) Most of the results are just a description of the study population. Needs further reporting on the findings of the EQ-5D data and VAS score.

Response: 

The EQ-5D results are reported in the results sections as mobility percentages, pain and discomfort, anxiety, and depression, and are also mentioned in Table 2.

Comment

15) The first time VAS score is introduced is the last sentence within the results. It is not described anywhere else prior to this. Needs to be included in the methodology. This data is lacking background and context.

Response: 

The VAS score has been added in methodology, with brief description. 

Comment

Discussion:

16) Some points of the scoring system(s) within the discussion need to be introduced earlier (Methodology/results). This will allow the reader to fully understand the results and what they mean before reaching the discussion.

Response: 

Thanks. As per your recommendation, we did this and added the scoring system and brief details in the methodology section.

Reviewer #3 comments and response from authors 

Comment

It is recommended to perform multivariate analysis (regression).

Response: 

Thanks for your recommendation, we performed the multivariate regression analysis, but the results were not so satisfied to include in the manuscript and the reasons are: 

1. We employed non-parametric test for our analysis, because of non-linear relationship and Regression analysis assumes a linear relationship between variables. 

2. For certain research objectives, such as exploratory or descriptive studies, regression analysis may not be applicable. Regression might not add much value to the study in certain circumstances; alternative methods of analysis may be more suited as we did in our manuscript to report the solid results. 

Comment

• Is the Urdu version of the EQ5D-3Lquestionnaire valid and reliable? Please provide a reference. And do all Afghan refugees in Pakistan have the ability to understand Urdu? Did you get the data through interviews?

Response: 

Yes, it is valid and reliable and we also mentioned the reference in methodology as reference number 22. By clicking that reference 22, everyone can easily find the EQ-5D-3L questionnaires in many different languages including Urdu. Yes, majority can speak and understand Urdu language. We did the data collection by asking them question only if they were having any problem in understating the EQ5D questionnaire. 

Comment

• In the method field, write the name of the analysis used.

Response: 

Thanks, it has been modified and mentioned the analysis used in the study.

Comment

• In the discussion section, write the strengths and weaknesses of the study.

• And keep in mind that your study does not have external validity because sampling was available and cannot be generalized to the Afghan refugee community in Pakistan.

Response: 

Thanks! We added the strengths and weaknesses section to the discussion part of the manuscript. We also mentioned that the results can’t be generalized to all Afghan refugees.

---

## [Editor Report · Decision Letter 2]

6 Jul 2023

Assessment of health-related quality of life among Afghan refugees in Quetta, Pakistan

PONE-D-21-26924R2

Dear Dr.Faiz Ullah Khan,

We’re pleased to inform you that your manuscript has been judged scientifically and is accepted for publication.

Kind regards,

AKM Alamgir, PhD

Academic Editor

PLOS ONE
---

## [Editor Report · Acceptance letter]

23 Jan 2024

PONE-D-21-26924R2 

PLOS ONE

Dear Dr. Khan, 

I'm pleased to inform you that your manuscript has been deemed suitable for publication in PLOS ONE. Congratulations! Your manuscript is now being handed over to our production team.

Kind regards, 

on behalf of

Dr AKM Alamgir 

Academic Editor

PLOS ONE